

# Fast simulation of detector effects in Rivet

**Andy Buckley[1⋆], Deepak Kar[2] and Karl Nordström[3,4]**

**1** School of Physics & Astronomy, University of Glasgow, United Kingdom
**2** School of Physics, University of the Witwatersrand, Johannesburg, South Africa
**3** Laboratoire de Physique Théorique et Hautes Énergies, Paris, France
**4** Nikhef, Amsterdam, Netherlands

⋆ andy.buckley@cern.ch

## Abstract

We describe the design and implementation of detector-bias emulation in the Rivet MC event analysis system. Implemented using C++ efficiency and kinematic smearing functors, it allows detector effects to be specified within an analysis routine, customised to the exact phase-space and reconstruction working points of the analysis. A set of standard detector functions for the physics objects of Runs 1 and 2 of the ATLAS and CMS experiments is also provided. Finally, as jet substructure is an important class of physics observable usually considered to require an explicit detector simulation, we demonstrate that a smearing approach, tuned to available substructure data and implemented in Rivet, can accurately reproduce jet-structure biases observed by ATLAS.



# 1   Introduction

The RIVET [1,2] framework is well established at the LHC and increasingly beyond as a standard toolkit and library of collider event analyses at "truth level", i.e. on events as they would be seen by a detector with ideal calibrations and infinite resolutions. In this it plays an important role for preservation and reinterpretation (e.g. in Monte Carlo event generator tuning) of experimental data from which detector effects have been *unfolded*.

Unfolding is an ill-posed problem [3] since there is no unique inversion of a convolution, and problematic because naïve maximum-likehood estimation methods such as direct inversion of a bin-migration matrix tend to be numerically ill-conditioned [4]. Demonstrating that an unfolding is well-understood, numerically stable, and demonstrates accuracy and robustness in closure and stress tests requires many extra studies even in nominally "simple" phase-spaces and observables. Uncertainties in the unfolding itself must also be estimated and included in the final result. As a result, taking a measured observable from the "reco" form directly reconstructed from a particle detector's outputs to a unfolded, detector-independent form usually adds very considerable time and effort. In experimental searches for new physics, the preference has thus almost universally been to perform the interpretation at detector level, both for speed and because searches often obtain model-sensitivity in observable bins whose statistics are too sparse for a stable unfolding but which can be interpreted without penalty using Poisson likelihoods.

The consequence is that anyone wishing to reinterpret BSM search data against a new signal model must provide a forward "folding" that adequately captures the biases and resolution degradations of the detector (and its reconstruction algorithms). In this paper we describe the design, implementation, and performance of such a fast detector-simulation system using the established RIVET analysis infrastructure, allowing detailed reproduction of analysis-specific detector effects for preservation of collider-experiment BSM search analyses in RIVET.

# 2   Design considerations

From the point of view of accuracy, the ideal form of "folding" would be to run the "full chain" simulation and reconstruction used by the experiments themselves/ This would require passing newly generated signal events to first the appropriate Geant 4 [5] geometry and material interaction models, then modelling the detector electronic response (digitization), and finally each experiment's reconstruction and analysis software that yields their calibrated physics objects. But none of this is publicly available, and even if it were, the typical CPU cost of multiple minutes per event renders it unfeasible to analysts other than the experimental collaborations themselves. In lieu of this option, the DELPHES [6] fast-simulation program has become the field's established fast simulation tool: it uses approximate experiment geometries and particle propagation models, combined with tabulated reconstruction efficiencies for different object

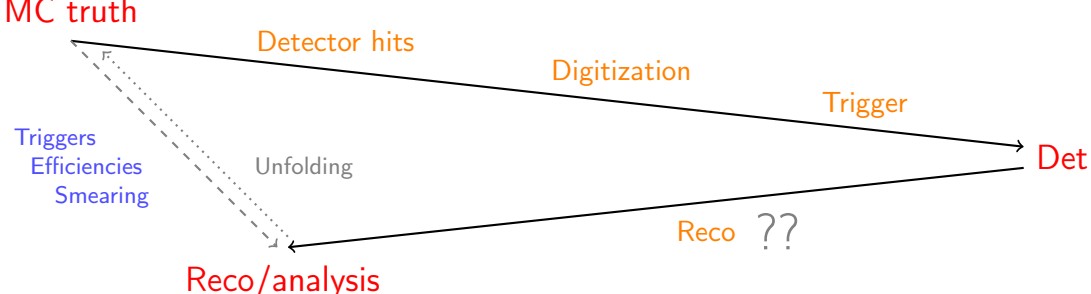

Figure 1: Smearing vs. explicit fast-simulation strategies compared, showing how a smearing approach short-circuits the little-known major effects of the detector and reconstruction processes individually, by instead parametrising the resultant relatively small effect of detector+reconstruction. The axes are arbitrary, but distances between points on the two trajectories represent the typical size of discrepancies between equivalent physics objects at the two stages of processing.

types, to produce approximate "reco-level" physics objects for analysis.

In RIVET, however, we have implemented a more lightweight approach based purely on use of effective transfer functions to map physics objects from truth-level to reco-level. In practice this largely means use of "smearing" (or convolution) of truth-level physics object kinematics by resolution functions, in conjunction with DELPHES-like tabulated or parametrised reconstruction efficiencies.

*A priori*, this simpler approach may appear less accurate than one including explicit detector modelling, but this is not necessarily the case. The motivation for our use of a smearing technique can be understood with the aid of Figure 1, which depicts the distortions of physics objects relative to their truth-level ideals by detector and reconstruction effects: large distances between points reflect large distinctions between equivalent objects. The solid black arrows show the two long legs of the approach of explicit simulation and reconstruction, the detector taking the physics objects to their maximum distance from truth level before explicitly reconstructing them as something much closer to truth (partially by construction – many calibrations make significant use of MC modelling, even when this is validated by data-driven methods). By contrast the shorter dashed grey arrow represents the net effect: it is this smaller effect that is amenable to the smearing+efficiency approximation[1]. We additionally note in the figure that, as an effective rather than explicit technique for mapping between process endpoints, it is analogous to the inverse of the unfolding procedure, or vice versa.

Indeed, there are several good reasons to believe that there is little or no practical accuracy gain in using an explicit fast-simulation rather than this transfer-function technique:

- Many of the processes on the long-trip legs are imperfectly known outside the experiments: the geometry and fast material maps will be significantly inaccurate compared to a full G4 model, and the digitization, trigger, and nearly all reconstruction details are only known internally. Imperfect modelling of large effects which are intended to largely cancel is a risky business.

- What is known about triggers and reconstruction/identification performance is published efficiency maps or resolution widths as functions of key kinematics such as $p_T$ and rapidity. These are hence equally accurate in smearing and explicit fast simulations, but the smearing formalism is directly expressed in these terms while explicit methods have to ensure that they are not spoiled by the approximate detector emulation machinery.

---

[1]From here onward we refer to the combination of smearing+efficiency as simply "smearing" for simplicity.

- Nuanced reconstruction effects, intimately dependent on detector and reconstruction details, are unlikely to be well modelled in either flavour of fast-sim, given the imperfect modelling in one case and its total absence in the other. In fact, the oft-cited example of a difficult effect is the modelling of detector-level jet substructure, to which we devote some effort via a smearing approach in Section 6.

In addition, we recall that the primary audience for detector simulation is BSM model reinterpretation of published data; no discovery will ever be claimed from such a study, although we hope and expect that interesting features discovered through reinterpretation analyses will prompt experiments to explicitly revisit data analyses with refined methods and BSM models. The accuracy required to make interesting observations is not high – due to the typically rapid fall-off of cross-sections and hence expected event yields with BSM theory mass-scales, reproduction within 10–20% of the original experiment's reco-level performance in key signal regions is frequently sufficient to place meaningful bounds on new physics [7–12]. The strength of fast simulation is its speed and public availability, and deficiencies in accuracy have to be very substantial before they impede practical usefulness.

In the following section we review the practical design and implementation of the RIVET smearing+efficiency fast simulation system, then move to its performance for various classes of physics object.

## 3 Implementation of RIVET fast-simulation

**Practical design aims**

Our first major aim for implementing detector-effect simulation in the RIVET framework was to make the interface "natural", i.e. to integrate well with the established particle-level programming interface, and to maintain compatibility with existing observable calculators ("projections" in RIVET jargon) which have received a great deal of design and debugging effort. RIVET's uptake among physicists being largely rooted in the API emphasising physics concepts, it was important that smearing machinery should not introduce significant noisy "boilerplate" code that would obscure the analysis ideas.

The smearing formalism lends itself particularly well to this approach, as it preserves an unambiguous link between particle-level and reco-level physics objects. An explicit detector simulation breaks this link due to the intermediate stage of detector hits, which in general can map multiple particle-level to multiple reco-level objects. Expressing detector effects as per-object transfer functions (and mappings to null reco-level ones, in the case of reconstruction inefficiencies) is also ideally compatible with detector performance metrics as published by experiments, meaning that the most accurate parametrisations can be applied directly rather than having to be constructed by cunning configuration of material and reconstruction components.

The second major requirement in our design was that detector simulation should in general be analysis-specific, rather than a monolithic detector simulation which is assumed to be appropriate for all analyses. The latter approach is the one taken by DELPHES, but in practice there are too many analysis-specific effects for the picture of a unique ATLAS or CMS detector emulation to pass muster: the MadAnalysis 5 [9,13] and CheckMATE [12] analysis databases, for example, provide different DELPHES detector configuration files for each analysis. Some of this is due to explicit choices in physics objects definition, e.g. jet radius and clustering measure [14], and $b$-jet tagging working points, while others arise from implicit, environmental factors such as the operating era of the detector (incorporating hardware upgrades, for example), and the version of reconstruction software used for the analysis. While detector performance papers publish generic performances for standard-candle processes in fairly inclusive phase spaces,

we are also aware that BSM analyses typically cut into extreme regions of phase space where detector performance may deviate from nominal: in our implementation we hence wanted custom, analysis-specific response functions to be used as easily as generic ones.

### Implementation

The implementation of smearing-based fast simulation in RIVET has been implemented using the existing `Projection` machinery, which automates caching of expensive computations. The plethora of existing particle-level projections which return lists of `Particle` objects – for example, `FinalState`, `TauFinder` and `UnstableParticles` – may be "wrapped" in the `SmearedParticle` class by passing to its constructor along with the relevant smearing and efficiency transfer functions. This interface is essentially interchangeable with the original, as it implements the same `ParticleFinder` interface as the particle-level projections it wraps. A similar `SmearedJets` wrapper projection is provided for wrapping any particle-level jets constructed with any jet algorithm and implements the same `JetAlg` interface as the particle-level finders, while a special `SmearedMET` wrapper is provided for measurements of missing (transverse) momentum using the particle-level `MissingMomentum` class.

The transfer functions provided to these projections' constructors can be any function or functor that supports a "callable" `std::function` interface – indeed, they are internally stored as `std::functions` in the smearing projections' implementations. In the cases of `SmearedParticles` and `SmearedJets`, these supplied functions take a `Particle`/`Jet` as argument, and return either a modified `Particle`/`Jet`, or a floating point number representing the reconstruction efficiency. As their operation is defined at a per-object rather than per-event level, smearing function implementations – both built-in standard forms, and custom ones supplied by analysis authors – avoid boilerplate code to loop over collection members and, in the case of efficiency functions, to perform C++'s rather arcane dance for their deletion.

Since the efficiency functions just return a probability rather than proactively carrying out their stochastic keep/reject mission (that being reserved for the containing projection), and smearing functions return a modified physics object, the functions can be composed to create multi-stage smearing and efficiency behaviours. Internally, both kinds of function are stored as combined smearing+efficiency `std::functions` which perform the map `Particle`/`Jet` → { `Particle`/`Jet`, `double` }, allowing an arbitrarily long chain of smearing and/or efficiency functions to be applied in a given order, should a very complex smearing procedure require such a general treatment.

The smearing and efficiency functions are most easily implemented as stateless functions, and can be written into the same C++ source file as the analysis logic, automatically binding the two together for analysis-preservation robustness. For maximum accuracy we encourage implementers to encode jet and particle reconstruction functions specific to their analysis, but for convenience and as prototype examples of how to make your own, implementations of detector smearings and efficiencies for ATLAS' and CMS' published Run 1 & Run 2 performances are provided in RIVET's `Rivet/Tools/Smearing.hh` and similar headers. These will be discussed in Section 4, and their performance demonstrated in Section 5.

In practice, for lepton reconstruction the efficiencies are a much more important effect than kinematic smearing and so are specified first in the `SmearedParticles` constructor's argument order, allowing a default identity map to operate for the smearing. The opposite is true for `SmearedJets`, since jets nearly always have 100% reconstruction efficiency but can suffer important kinematic degradation – although we note that purely jet-and-MET BSM searches can in fact be implemented fairly robustly with no detector simulation at all [15, 16]. The `SmearedJets` constructor also accepts optional $b$-tag and $c$-tag efficiency/mistag rate functions, which like the jet efficiency functors map `Jet` → `double`. As tagging rates are frequently more sensitive than either the whole-jet efficiency or its kinematic smearing, jet rerconstruction can

be specified as tagging rates alone, with the whole-jet defaulting to identity smearing with 100% efficiency.

A benefit of the programmatic approach to specifying transfer functions, as opposed to tabulation, is that the full range of C++ features are available to be used. For example, the jet $b$-tagging efficiencies may be expressed very compactly for jets whose truth-flavour is $b$, $c$ or light, via the chained construction

`return j.bTag()? 0.7 : j.cTag()? 0.1 : 0.01,`

which gives a 70% $b$-tag efficiency, and mistag rejection rates of 10 and 100 for $c$- and light-jets respectively. Alternatively, a standard functor struct is available for those queasy about such syntactic trickery: `JET_BTAG_EFFS(0.7, 0.1, 0.01)`. Many more standard functors and filtering tools for use in transfer function and BSM analysis implementation are provided, and even "modern C++" features like lambda functions may be used seamlessly. Since tabulation is sometimes required, functions have been provided to assist with 1D and 2D binned efficiency and resolution lookup (in $\eta$ and $p_T$, for example).

As all `Projection` objects must be comparable to make RIVET's caching system work, the comparison logic treats the contained functions as equivalent if they can be resolved into function pointers *and* those pointers are equivalent between `SmearedParticles`/`Jets` instances; if the resolution to function pointers cannot be made, the conservative option of recalculation is taken. This can perhaps be further optimised, but the particle-level caching of e.g. expensive jet clustering will in all cases continue to work within the smearing-function wrappers.

## 4   Standard object smearing and efficiencies

Analysis-specific is the ideal, but generic functions are provided in the RIVET package, based on public calibration and performance papers from the ATLAS and CMS experiments as documented in this section.

These standard functions are located in the `Rivet/Tools/SmearingFunctions.hh` header, and have all-caps names in a structured form that encodes the experiment, collider run, type of function (`SMEAR` or `EFF`), and details such as the name of the efficiency working point; an illustrative example is `ELECTRON_IDEFF_ATLAS_RUN2_TIGHT(const Particle& e)`. Helper functions are provided for distribution sampling e.g. `rand01()` and `randnorm()`, bin lookup in tabulated efficiency maps e.g. `binIndex(double, vector<double>)`, and momentum smearing functions that handle details like ensuring energy- and $p_T$-positivity after smearing, e.g. `P4_SMEAR_PT_GAUSS(FourMomentum&, double)`. These are located in other `*Smearing *.hh` headers in the `Rivet/Tools` folder, and may be used to implement user-supplied detector functions specific to an analysis, as well as the generic LHC-experiment ones.

### 4.1   Jets

Smearing is by far the dominant effect for jets since, unlike for leptons, jet triggers are typically used in their fully-efficient regime. Accordingly no standard jet efficiency functions are supplied (the default being 100%), but a standard jet resolution function is provided. The ATLAS Run 1 jet energy resolution has been implemented to match the result of Ref [17], as a parametrisation in $p_T$ alone. Gaussian smearing of the jet energy resolution is applied to the 3-vector components, while preserving the jet mass. A separate jet mass smearing could be applied in custom smearing functions, in either ordering, using the ability to chain multiple smearing functions. The ATLAS Run 2 and CMS Runs 1 and 2 resolution functions are copies of this, matching the equivalent ATLAS/CMS jet resolutions used in DELPHES.

## 4.2 Jet flavour tagging

As the RIVET smearing system has full access to the truth-level particle constituents of all jets, arbitrarily detailed truth-level parametrisation of tagging behaviour is possible if needed. The tagging is based, following the general RIVET physics philosophy, on the truth-tagging function based on weakly-decaying $b$ (and $c$) hadrons ghost-associated [18] to the final-state jet[2].

ATLAS requires the truth jet to contain a $b$-hadron with $p_T > 5$ GeV, while CMS accepts any $b$-hadron/quark in the jet. These requirements are enforced by the tagging efficiency functions as well as in any user calls to e.g. `myjet.bTagged(Cuts::pT > 5*GeV)`. The function `JET_BTAG_ATLAS_RUN1` uses the DELPHES parametrisation, which drops asymptotically to zero efficiency. Run 2 ATLAS tagging efficiency/mistag functions do not have this behaviour, and simply return the official mistag and efficiency rates for the 77% working point of the MV2c10 and MV2c20 taggers.

Since flavour tagging is sensitive to the detailed working points used in an analysis, as well as the calibration period in which the analysis was made, it is strongly recommended that the specific efficiency and mistag rates given in each paper be used. A `JET_BTAG_EFFS` struct is provided to make this easily specifiable without resorting to "noisier" C++ lambda functions or similar: this can be passed to the `SmearedJets` constructor as a tagging-efficiency functor.

## 4.3 Tracks

The ATLAS and CMS tracking performances are based on the DELPHES efficiency parametrisations, which have distinct forms for electrons, muons, and charged hadrons, and are tabulated within rough $p_T$ and $\eta$ bins for each. Where DELPHES uses three separate functional blocks to define these efficiency functions, with the dispatch by particle ID hard-coded, the RIVET implementation provides the same functionality via a single user-defined function such as `TRF_EFF_ATLAS_RUN2`. Such functions can use any property of the passed `Particle` object to influence the choice of parametrisation. Following DELPHES, the two experiments currently use the same efficiency map, and the Run 1 and Run 2 behaviours are currently identical.

## 4.4 Electrons

ATLAS electron performance is based on tabulated $(|\eta|, p_T)$ efficiencies and parametrised resolutions, with the total efficiency split into a reconstruction and an ID component, with the usual loose, medium, and tight working points specific to the ID part. Run 1 efficiencies are based on the DELPHES steering cards, and the Run 2 efficiencies were extracted from the relevant ATLAS performance notes and papers [19–21]. The CMS efficiencies do not have the reco/ID split.

For both experiments, the momentum resolutions are based on the DELPHES parametrisations. These take the form $\sigma(E) = \sqrt{aE^2 + bE + c}$ in ATLAS, for electron energy $E$, and tabulated coefficients $a, b, c$ as functions of $(\eta, p_T)$. CMS' equivalent smearing uses the form $\sigma(E) = \sqrt{a^2 + b^2 E^2}$ with $a, b$ tabulated in $|\eta|$ only. These resolutions in both cases are used as the width of a Gaussian smearing of the energy, with an energy-positivity enforcement and no directional modification.

In practice, hadronic jets can fake a reconstructed electron. As the types of `Jet` and `Particle` are distinct in the current implementation, this phenomenon is not automatically reproduced –

---

[2]Ghost-association is a method for jet-area computation and for tagging of jets with unstable event objects such as $b$-hadrons. Unlike $\Delta R$ matching, ghost-association is integrated with the full jet clustering sequence, and makes use of the infra-red safety properties of actively used jet algorithms to not bias the jet kinematics through double-counting. The method involves scaling the tag-object 4-momenta to negligibly small "ghost" values, then clustering them along with the usual final-state event objects.

and anyway no fake-rate is currently published, but it is easy to apply a custom `SmearedJets` efficiency function as a model of accidental jet reconstruction as a photon or electron

### 4.5 Muons

As for electrons, muon efficiency is decomposed into reconstruction and ID components. However, the efficiency variation between loose, medium, and tight ID working points is relatively small and so only the medium working points are currently encoded, for Run 1 by repetition of the efficiency tabulation used by DELPHES, and for Run 2 by tabulation of the efficiency performance plots as functions of $p_T$ from Ref. [22]. The CMS muon efficiency is again treated monolithically, using the $\epsilon \propto \exp(a - b p_T)$ parametrised form from DELPHES.

The kinematic smearing for ATLAS is implemented for both Runs 1 and 2 as Gaussian smearing of the $p_T$ via relative $p_T$ resolutions $\sigma(p_T)/p_T$ tabulated as a function of $(|\eta|, p_T)$ [23]. CMS's equivalent muon smearing is based on the DELPHES parametrisation, using Gaussian $p_T$ smearing with relative resolution $\sigma(p_T)/p_T = \sqrt{a^2 + b^2 p_T^2}$ with $a, b$ tabulated in $|\eta|$ only.

### 4.6 Taus

Hadronic tau identification is difficult experimentally, and accordingly leads to a complex implementation in terms of truth objects. The efficiency parametrisations are based on tabulations as usual, but separated into different categories based on the number of charged hadrons ("prongs") in the decay, and the sum of hadronic visible $p_T$ from generator-stable particles. The ID functions are implemented both as `SmearedParticles` functors for application to truth-tau particles, and as `SmearedJets` functors for jets. In practice, the experimental treatment of hadronic tau ID is more similar to jet flavour-tagging than electron and muon reconstruction, and tau reconstruction is likely to evolve toward the jet and tag-efficiency/mis-ID appraoch in future releases.

The ATLAS Run 1 tau medium working-point tau ID efficiencies are provided, based on Ref. [24], and the Run 2 medium working-point efficiencies from Ref. [25]. Following DELPHES 3.3.2, the CMS tau efficiency is a fixed 60% in DELPHES 3.3.2. The tau smearing for both ATLAS and CMS, and for both particle-based and jet-based tau reconstruction, is the same as for jets in the relevant collider runs.

### 4.7 Photons

ATLAS photon construction is based on the converted photon efficiencies from Ref. [26] for Run 1, and from Ref. [27] for Run 2, in both cases tabulated as functions of $(\eta, p_T)$. CMS's photon efficiencies are based on the simple form from DELPHES. As for electrons, which similarly are reconstructed based mainly on ECAL information, no built-in mechanism is provided for jets faked by photons, and again the fake is rate not published, but a `SmearedJets` efficiency may be used to emulate jet misreconstruction as a photon or electron in cases where the fake rate is important; electrons faking photons (and vice versa) can be built directly into the particle efficiency functions cf. jet mistagging. No kinematic smearing is currently applied to photons.

### 4.8 Missing transverse momentum

Since MET is measured and calibrated using all the visible objects in the event, the smearing of missing $E_T$ by `SmearedMET` requires not just the truth-level MET vector, but also a measure of whole-event activity. Currently this is achieved by passing the MET vector and the event scalar $\sum E_T$ (SET), as used by performance-characterisation papers; this interface may evolve, e.g. to be passed the entire RIVET `Event` in a later iteration to allow arbitrary inputs to the calibration as smearing functions become more refined.

Unlike all other objects, there is no efficiency to be calculated for MET, only kinematic mismeasurement. The ATLAS MET smearing is based for Run 1 on the performance figures measured in Ref. [28]. This uses a linearity offset as a function of $\not{E}_T$, then calculates a MET resolution via the parametrisation $\sigma(\not{E}_T) = a\sqrt{\sum E_T}$; this is then used for Gaussian smearing of the modulus of the 2D MET vector, with a positivity constraint. The CMS smearing is similar, but distinct $x$ and $y$ MET resolutions are computed based on the $\sum E_T$, based on the approach described in Refs. [29] and [30] for Run 1 and Run 2 respectively.

## 5 Validation of smearing performance

In the absence of significant truth-level vs. reconstruction-level MC data from the LHC experiments with which to validate an external fast-simulation package, we will now compare the performance of the RIVET smearing approach to the more explicit approach in the form of DELPHES 3.4.2. Neither code is guaranteed to be the more correct, but as both have been tested with success in e.g. Les Houches comparisons of BSM analysis recastings [15][3] it is expected that their performance on basic event observables should be comparable and perhaps indicate the current acceptable range of fast-simulation uncertainty.

To provide a source of most relevant physics objects, we use two event samples: 100k events of inclusive top-quark and $t\bar{t}$ production by Pythia 8.235, with all $W$ bosons decaying to $e/\mu$; and 100k of $t\bar{t}\gamma$ with $W$ decays to tau leptons, simulated by LO MadGraph 2.6.7 [31, 32] and Pythia 8 [33, 34]. These samples provide a high-statistics source of ($b$-)jets, stable leptons, prompt photons, and taus, and a busy detector environment in which to test the effects of object isolation and overlap removal.

The same truth-level events were passed to RIVET and to DELPHES, using ATLAS Run 2 detector configurations, and analysed with equivalent simple analysis codes which plot the multiplicity, $p_T$, and $|\eta|$ of each class of object ($e^\pm$, $\mu^\pm$, jets, $b$-jets, and missing transverse energy), with the kinematic variables also recorded specifically for the leading (highest-$p_T$) objects in each event. Since we are interested in both the inclusive modelling of event features, and the more selective view in which jets and leptons have been isolated from one another by further analysis-level cuts, the RIVET analysis and DELPHES simulation step were run both with and without isolation, with the RIVET version written to apply equivalent "relative" isolation cuts to those built into DELPHES.

The results of this simulation for stable charged leptons are shown in Figure 2, with electrons in the left column, muons on the right, and multiplicity, $p_T$, and $|\eta|$ distributions in the rows from top to bottom. A $p_T$ requirement of 10 GeV was applied to both lepton flavours, at the relevant {truth,reco} level of simulation. As expected, there is a smooth "staircase" decrease in rate for inclusive lepton multiplicities, with RIVET and DELPHES seen to agree closely for up to 5 leptons for both flavours – unsurprising as both are based on similar efficiency tables. For the isolated leptons, a sudden drop in rate is seen above two electrons or muons per event, in addition to a larger effect of isolation in the 2-leptons bins than in the 1-lepton bins: this reflects the true event topology and the effectiveness of the isolation cut in removing high-$p_T$ leptons from jet fragmentation. Indeed, this effect is seen also in truth-level isolation. Again, RIVET and DELPHES concur in their efficiencies for up to two isolated leptons. The $p_T$ and $\eta$ plots also show general agreement between RIVET and DELPHES, with disagreements typically of 5% and inflating in some regions e.g. low-$p_T$ electrons by $\sim$ 10%. Isolation again has a similar effect, mirrored in the $p_T$ distributions by the truth-isolation curve – the similarity of truth and reconstruction shapes indicate that main detector effect is again the lepton efficiencies. These are again evident in the $|\eta|$ distributions, where the rough tabulation of available efficiencies is

---

[3]Using MadAnalysis and CheckMATE custom steering cards for DELPHES.

visible in the step shape in the RIVET and DELPHES ratios with respect to the truth. The geometric acceptance of the detector systems is also evident here, although with some discrepancies in the maximum $|\eta|$: RIVET cuts off electron reconstruction at the tracker $|\eta| < 2.5$ and muon system $|\eta| < 2.7$, while DELPHES has steps at $|\eta| < 2.5$ followed by longer tails somehow out to $|\eta| = 4$ and beyond.

Figure 3 contains a similar set of multiplicity, $p_T$, and $|\eta|$ distributions for inclusive jets on the left and $b$-jets on the right, both with a $p_T > 20$ GeV requirement. These distributions highlight issues with DELPHES unisolated jets and with $b$-tag reconstruction from MC records without explicit $b$-quark content (ATLAS uses hadron-based $b$-tagging). The inclusive jet multiplicity distribution from RIVET closely matches the truth jet multiplicity, as expected since jet reconstruction rates are effectively 100% and the main reconstruction effect for jets is momentum smearing that leads to migrations between bins or out of $p_T$ or $\eta$ acceptance. The DELPHES inclusive jet multiplicity distribution is shifted to lower values than the truth and RIVET rates, suggesting significant loss of inclusive jets; this shifts toward much closer agreement between truth, RIVET and DELPHES when jet isolation criteria are applied to all three. The $b$-jet multiplicities obviously cannot be compared for DELPHES, but fit the expected migration pattern in RIVET for the 70% $b$-tag working point used in both simulations. Both the inclusive jet and $b$-jet $\eta$ distributions show almost no shape dependence in RIVET– just the expected 100% and $\sim$ 70% efficiencies respectively – but DELPHES introduces significant shaping with an apparent depletion of jets within the tracker acceptance being somehow reversed by application of the isolation cut. Some isolation effect is seen for the few $b$-jets, unlike in RIVET where the $b$-jets see little effect due to isolation from already-isolated leptons, but notably $b$-jets are reconstructed outside the DELPHES tracker. Finally a DELPHES excess in low-$p_T$ jets with respect to truth and RIVET is resolved by application of the isolation criteria.

The last distribution in Figure 3 is the missing transverse energy (MET). Discrepancies of around 10-20% are seen between RIVET and DELPHES in both low and high values of this distribution, which is well populated across the spectrum due to the presence of no-neutrino, 1-neutrino, and 2-neutrino signal processes. The RIVET MET distribution is closer to the truth everywhere, especially at large values of MET where the relative calibration uncertainty reduces as the MET is constructed from the imbalances of more central and more collimated high-$p_T$ visible objects. Both RIVET and DELPHES see a 20-40% enhancement of "soft MET" at values below 5 GeV, but the DELPHES deviation from the truth proceeds linearly until a crossover point at around 60 GeV, while the RIVET modelling undershoots the truth from 10-40 GeV and is then in agreement with the truth within a few percent. This disagreement is likely to be significant for reinterpretations of BSM signals with large true MET of 100 GeV or more.

Finally, Figure 4 shows the multiplicity, $|\eta|$, and $p_T$ distributions for high-$p_T$ photons and tau leptons. While general trends typically agree, such as the relative effects of misreconstruction and isolation/overlap removal, significant deviations between DELPHES and RIVET are again visible. Some are due to choices such as RIVET's inclusion of the crack between the ATLAS barrel and endcap calorimeters in efficiency tabulations, seen in the $|\eta|$ plot, others such as the photon efficiency (which for DELPHES is around ten times lower with respect to the truth than RIVET's version, the latter being compatible with the $\sim$ 80–90% from the performance paper [27]) and DELPHES' higher tau multiplicities (flat between one and two taus, and higher for isolated than unisolated) have less obvious origin.

Overall, we can conclude that there is still significant uncertainty in the results of LHC fast-simulation codes based on published detector performance numbers, while the broad features of smearing and explicit fast-simulation are similar. This further motivates the availability of multiple fast, public detector-simulation codes, as well as highlighting the importance of experiments' providing analysis-specific resolution and efficiency characterisations, to reduce such downstream modelling uncertainties.



Figure 2: Electron and muon performance from Pythia8 LHC $t\bar{t}$ production at truth level, and after processing through the standard ATLAS detector emulation routines in RIVET and DELPHES. Electron observables are shown in the left column, and muons in the right; the observables are multiplicity, $p_T$ distribution, and $|\eta|$ distribution from top to bottom.

# 6 Jet substructure smearing

The smearing of overall jet kinematic observables which are insensitive to spatial correlations between constituents like the $p_T$, energy, and direction can be approximated without modelling the finite spatial resolution in $\eta$–$\phi$ space of the calorimeters themselves. This is because the

Figure 3: Jet and MET performance from Pythia8 LHC $t\bar{t}$ production at truth level, and after processing through the standard ATLAS detector emulation routines in RIVET and DELPHES. Inclusive jet observables are shown in the left column, and $b$-jets in the right with the exception of the bottom row which shows the MET distribution on the RHS. For the jet objects, the observables shown are multiplicity, $p_T$ distribution, and $|\eta|$ distribution from top to bottom.

total detector response effectively is a sum of the response of a number of individual energy deposits in the calorimeter, which each respond by a convolution of a number of physical effects which can be approximated as Gaussian by the central limit theorem. Then if the observable we are interested in only depends on summing the response in these different components



Figure 4: Photon and tau performance from MG5+Pythia8 LHC $t\bar{t}\gamma$ production at truth level, and after processing through the standard ATLAS detector emulation routines in RIVET and DELPHES. Photon observables are shown in the left column, and tau one in the right. The observables shown are multiplicity, $p_T$ distribution, and $|\eta|$ distribution from top to bottom.

together, the smearing of the observable will also be Gaussian in nature.

However observables like the jet mass which are sensitive to spatial correlations between components of the jet can not be treated like this. The most obvious issue is the finite spatial resolution of the detector, which for example suggests any jet which is narrow enough will be reconstructed as massless, regardless of actual mass. It is well-known that it is therefore

necessary to model the finite detector resolution in order to get a realistic detector response for the jet mass. This can be done by for example dividing up the $\eta$–$\phi$ space into realistic calorimeter cells, summing up the energies of the particles entering each (potentially with a species-dependent factor), and clustering jets with these pseudo-calorimeter cells as inputs rather than the raw particles[4].

For observables which are designed to be extremely sensitive to the spatial distribution of the radiation inside the jet it is not clear that even such a treatment is sufficient. In particular the assumption that all energy of a particle is deposited in the cell it enters, even if it is on the border between two cells, does not correspond to a realistic modeling of how calorimeters measure the energy of particles entering them. As the shower of radiation produced when a particle enters a calorimeter is spatially distributed, it is well-known that a hard and narrow jet which enters only a single cell will in general have some energy bleed out into the neighbouring cells. This means the reconstructed jet will appear wider than the particle-level jet. This effect can be expected to have a significant impact on jet substructure observables which are designed to look for subtle spatial correlations between jet constituents such as $N$-subjettiness [35]. It is therefore necessary to develop a method to simulate this effect in a fast and transparent manner, to investigate whether or not it is a relevant effect for realistic studies.

We will model it by smearing the direction of the particles entering the calorimeter cells before summing up the energy in each, rather than by smearing the calorimeter cells themselves. This allows us to take into account that a particle which enters a cell right on the border to another cell is more likely to deposit energy in this neighbouring cell rather than the one on the opposite side. It also allows us to make this 'directional smearing' a function of the incoming individual particle $p_T$ or energy, as we physically expect the significance of this effect to vary with the hardness of individual particles rather than the total energy entering a cell. In general we expect harder particles to have better directional resolution but also leak more energy into neighbouring cells. However as hard hadronic particles will tend to be accompanied by collinear softer hadronic particles (as very generally expected from jet formation), simply smearing the direction of softer particles will create a similar leaking effect without altering the overall kinematics of the jet (as could happen if hard particles are significantly smeared).

Following the general form of the detector response we expect, we will smear the $\phi$ and $\eta$ directions of incoming particles by a Gaussian with a mean of zero, and standard deviation given by[5]

$$\sigma(p_T) = \frac{A}{1 + e^{(p_T - B)/C}}, \tag{1}$$

where $A$, $B$, $C$ are constants which we fit to data. The same values are used for smearing both $\phi$ and $\eta$, which should be a reasonable approximation in the central region of the detector.

## 6.1 ATLAS Run 1 jet substructure

In order to fit the three constants to data we use the validated RIVET implementation of the ATLAS jet substructure analysis presented in Ref. [36]. This is the only public study to present both reco-level and unfolded measurements of several jet substructure observables with identical binning – although still only a small fraction of the total set of unfolded observables in that paper. We may hence directly extract detector transfer functions for these observables. A direct fit of $A$, $B$, $C$ to reco-level observables would risk correcting for deficiencies in the parton shower Monte Carlo description of the events rather than describing the detector response;

---

[4]We will throughout model the pseudo-calorimeter cells as massless momenta directed towards the centroids of the cell.

[5]The sigmoid parameterisation is fairly general for the type of behaviour we expect and can approximate other alternatives like a softmax parameterisation.

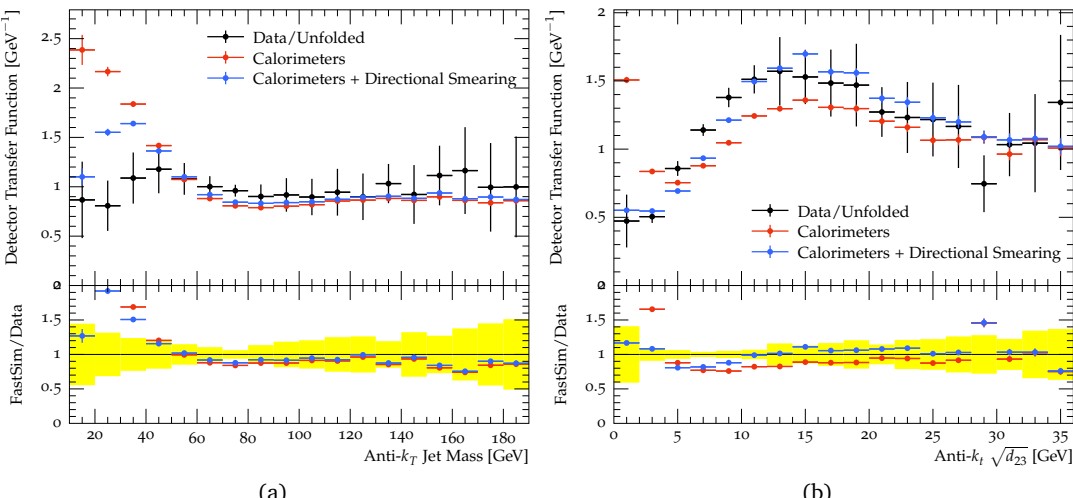

Figure 5: The detector transfer function extracted from data by dividing the detector-level distributions by the unfolded distributions, and two fast detector simulations with and without directional smearing for the $R = 1.0$ anti-$k_t$ jet mass and splitting scale $\sqrt{d_{23}}$ as measured in a 7 TeV inclusive dijet sample with $p_{T,j} \in [300, 400]$ GeV in Ref. [36].

we hence chose instead to fit to the ratios of reco-level to unfolded distributions, reducing the dependence on the quality of MC physics modelling.

The fit was performed using Professor [37] to minimise the $\chi^2$ between the detector transfer functions extracted from the measurements reported by ATLAS and the detector transfer functions for the same measurements as obtained by using a simulated sample generated using Pythia8 [33, 34] event generator and Monash tune [38] (although the exact generator setup used in generating these events is unimportant as the ratios between generator and and reco-level distributions are used) and the validated RIVET analysis. The best fit obtained was

$$A = 0.045, \quad B = 31 \text{ GeV}, \quad C = 9.7 \text{ GeV}. \tag{2}$$

As the size of the calorimeter cells in the central detector is approximately $0.1 \times 0.1$ in $\eta$–$\phi$, the size of $A$ suggests a very modest effect which is effectively cut off for hard particles with $p_T \gtrsim 40$ GeV[6]. Significant soft radiation is at most moved into a neighbouring calorimeter cell and so does not significantly affect IRC-safe observables. If it is clustered into a jet, the directional smearing effectively mimics energy leaking between calorimeter cells.

To illustrate the effect on various jet substructure observables we show a comparison of the detector transfer function as measured in data, as modeled by calorimeter clustering and energy resolution smearing, and as modeled by calorimeter clustering and energy resolution smearing after directional smearing of the particles using the parameterisation described above in Figures 5, 6.

Figure 5 illustrates that standard calorimeter clustering does a reasonable job of modelling the detector transfer function everywhere except for the low mass and low splitting scale bins for the $R = 1.0$ anti-$k_t$ jet mass and $\sqrt{d_{23}}$ observables [39]. The addition of directional smearing does not significantly improve the agreement except for these bins.

Figure 6 on the other hand demonstrates that the standard calorimeter clustering breaks down when looking at more complex jet substructure observables like $N$-subjettiness. Here the

---

[6]We are here guilty of some *post hoc* rationalisation: the form of constituent smearing function was iterated based on first experience with this fit.

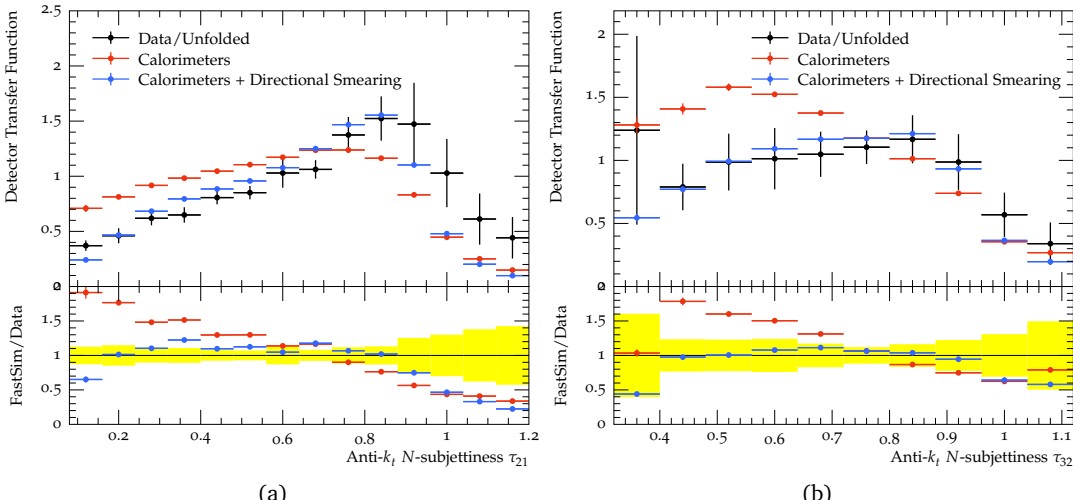

Figure 6: The detector transfer function extracted from data by dividing the detector-level distributions by the unfolded distributions, and two fast detector simulations with and without directional smearing for the $R = 1.0$ anti-$k_t$ jet $N$-subjettiness observables $\tau_{21}$, $\tau_{32}$ as measured in a 7 TeV inclusive dijet sample with $p_{T,j} \in [300, 400]$ GeV in Ref. [36].

addition of our very modest directional smearing of soft particles significantly improves both the qualitative and quantitative agreement between the detector transfer functions from data and our fast detector simulation.

## 6.2 ATLAS Run 2 jet substructure

The recent release of more recent ATLAS jet structure observables [40], with data available at both reconstruction and detector-corrected levels, allowed us to also test the performance of our tuned "ATLAS 2011" substructure smearing against new data. We found this to be unsuccessful, although whether this is due to the sensitivity of the procedure or due to evolution of ATLAS jet calibration remains unclear. However, a fit was again possible, following similar methodology. Out of the observables measured, LHA [41, 42], $D_2$ [43], and ECF2$^{\mathrm{norm}}$ [44] substructure variables using trimming [45] were used. These choices are somewhat arbitrary, but intended to cover substructure variables with different sensitivity. Again Pythia8 generator with Monash tune was used. It was discovered, however, that the "calorimeter granularity" segmentation built into the earlier approach was a limiting factor in the smearing, introducing a minimum degree of cluster smearing which was greater than that seen in ATLAS Run 2 jets. Removal of the segmentation, and tuning the same functional form for smearing as before, but applied to visible final-state particles rather than aggregated pseudoclusters, was found to again give a good description of substructure distributions. The tuned parameters for the unsegmented smearing extracted using reco/unfolded data ratios for the above mentioned substructure variables were

$$A = 0.028, \quad B = 25 \text{ GeV}, \quad C = 0.1 \text{ GeV}, \tag{3}$$

and a cluster energy resolution of $\Delta E = 1\%$, producing the distributions in Figure 7. Here the most sensitive variables the peak regions of the LHA and $D_2$ variables, which are generally well handled by the substructure smearing ansatz, except for the first bin of $D_2$. The LHA ratio distribution of observable is closely reproduced by this form of substructure smearing, as is the

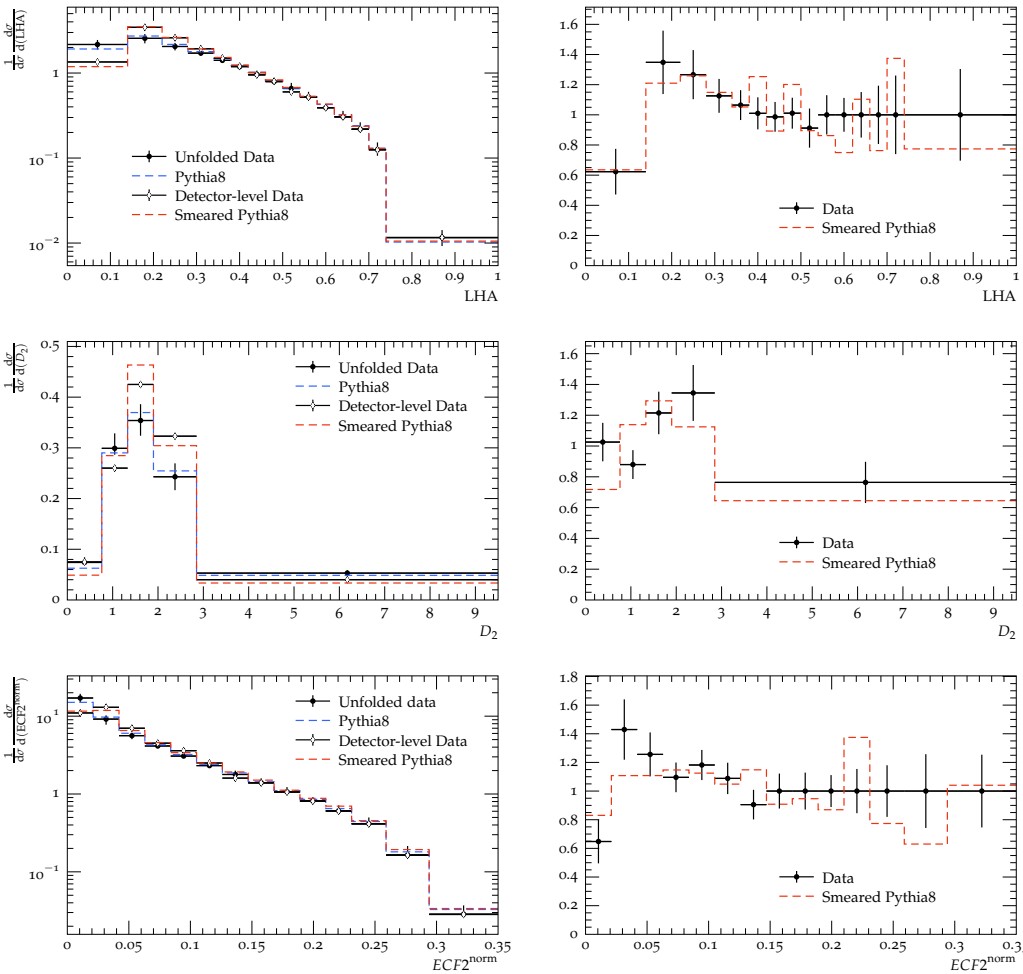

Figure 7: The effect of tuned jet constituent smearing without a calorimeter granularity emulation, on reconstruction-level data in Ref. [40]. The LHA, $D_2$, and $ECF2^{\text{norm}}$ observables are shown respectively in the rows from top to bottom, with the observed data compared to truth & smeared predictions in the left column, and the bin-by-bin transfer functions used for tuning in the right column.

relative reconstruction-invariance of the $ECF2^{\text{norm}}$. It must be noted that our parameterisation reproduces the interesting feature observed in the first two bins of the $ECF2^{\text{norm}}$ variable.

Taken together these results suggest that the energy leaking effect we model as directional smearing of soft hadronic particles can have a significant effect on the qualitative and quantitive agreement of the detector transfer functions for jet substructure observables as measured in data and obtained through fast detector simulation. Since there are currently a very limited number of public measurements which allow the detector transfer functions for jet substructure observables to be obtained from data in this manner[7] we are currently unable to study how well our parameterisation of this effect generalises. In particular it would be important to investigate how well it generalises to new signatures (for example boosted top quarks) and jet $p_T$ ranges. Input from the experiments is crucial for this to happen.

On the phenomenological side we invite researchers to investigate our jet substructure smearing in studies of for example top tagging, where the detector simulation arguably is one of the most significant systematic differences between phenomenological studies and experimental

---

[7]The necessary inputs are detector-level and unfolded distributions with identical binning.

studies of data.

## 7 Conclusions

We have described the implementation of a detector-bias emulation system in the RIVET MC collider event analysis system, implemented using a combination of efficiency and kinematic smearing functors in C++. The system makes use of modern C++ features to allow storing and combination of arbitrary function objects, allowing detector behaviours to be customised specifically to the phase-space and reconstruction working points of each analysis. In addition, a set of standard detector functions for the different physics objects in Run 1 and Run 2 of the ATLAS and CMS experiments has been implemented, based on a combination of parametrisations used in the DELPHES simulator, and by custom extractions from Run 2 detector performance publications. Comparison with DELPHES shows general agreement in the directions of detector bias, but significant differences that highlight the challenges of "external" detector emulation. As a final demonstration of the new functionality, we have demonstrated that applying RIVET smearing functions to the constituents of hadronic jets can reproduce detector biasing of jet substructure variables, given detector-response data for tuning the smearing functions. This exercise highlights that jet substructure modelling can be achieved without a full detector simulation, and hence a need for equivalent experimental observables to be published at both reconstructed and unfolded levels, to allow derivation of such response functions.

## Acknowledgements

AB is grateful to the Royal Society for a University Research Fellowship (grant UF160548). This work has received funding from the European Union's Horizon 2020 research and innovation programme as part of the Marie Skłodowska-Curie Innovative Training Network MCnetITN3 (grant agreement no. 722104). DK is supported by the National Research Foundation of South Africa (Grant Number: 118515).

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
