# Peer review of "Fast simulation of detector effects in Rivet"

_SciPost Physics, doi:SciPost Phys. 8, 025 (2020)_

## Round 1 · Referee Report · Jonathan Butterworth (Referee 1) · 2019-10-7

Strengths

  1. Presents a useful, widely-applicable and well-designed software tool for particle physics
  2. Presents a reasonable selection of demonstration results that the tool is performant
  3. Presents enough of a guide that users should be able make use of the tool and adapt it to their needs.

Weaknesses

  1. Does not, in itself, contain original physics results (though this is not the intention, this is an enabling paper).
  2. Contains a few unsubstantiated claims (see report).

Report

The smearing tools presented should broaden the usefulness of the (already widely used) Rivet library to allow the inclusion of detector-level/reconstruction-level results which have not been corrected/unfolded for detector resolution and efficiency.

In section 1, the claim is made that "sound unfolding" adds very considerable time and effort. It is not really clear that this needs to intrinsically be the case. If the reco-level distributions are well enough understood for a publication (including systematic uncertainties etc) then the final unfolding step can be relatively trivial. I think the more serious problem is that reco-level distributions are often *not* this well understood, but are nevertheless published. Once can always imagine "pathological" exotic cases where the unfolding is unreliable, but in such cases the parameterised approach used here (and in some cases even the full detector simulation and/or the detector calibrations!) would also be unreliable.

(The desire of searches to use sparsely-populated bins does seem to be an intrinsic limitation on unfolding, however.)

In Section 2 the authors' approach is described as "a priori less accurate" than a DELPHES like detector model. I am actually convinced by the authors that in fact their approach is a priori *more* accurate in many cases, since the functions are tailored to specific analyses. Any generic detector simulation based on efficiency maps must surely contain compromises which will not be equally accurate for all event topologies?

The authors also state that "most calibrations are highly dependent on MC modelling". They should clarify what they mean by this. I don't think calibrations should be (or are) in the end highly dependent on the event generation, since they are validated using in situ measurements. However, they are dependent on the MC model of the detector (which is validated in various ways using data).

In section 2, what justification is there for saying the 10-20% accuracy would lead to "conservative" bounds? Couldn't they just as well be over-aggressive, depending upon the sign of the error made?

In section 3 the authors mentuon "unconscious factors". I don't think unconscious is the right word here?

Section 4.1 It seems odd, though probably justifiable, that Energy smearing is applied to momentum and that the mass is left unchanged. I presume the energy is recalculated so the result is a valid four vector? If the mass is to be then subsequently smeared, would the energy again be recalculated automatically? Some clarification would help, I think.

Section 4.2 "ghost-associated" needs description/reference

4.8 Jets have no efficiency calculated either? So it is not just MET...

Section 5 was the 10GeV pT applied to truth or smeared value?

Requested changes

  1. Please address the comments in the report, some will imply changes.

Minor things:

  1. Section 3 double reference for MadAnalysis should be removed. The word "implemention"appears too often in one sentence.

  2. Page numbers would be nice.

  • validity: high
  • significance: good
  • originality: good
  • clarity: high
  • formatting: good
  • grammar: good

Author:  Andy Buckley  on 2019-10-15  [id 628]

(in reply to Report 1 by Jonathan Butterworth on 2019-10-07)
Category:
answer to question

Thank you for the very helpful and generally positive comments. We found the suggesetd clarifications useful and have included them in the next draft while we await further referee comments. Here are our point-by-point replies:

In section 1, the claim is made that "sound unfolding" adds very considerable time and effort. It is not really clear that this needs to intrinsically be the case. If the reco-level distributions are well enough understood for a publication (including systematic uncertainties etc) then the final unfolding step can be relatively trivial. I think the more serious problem is that reco-level distributions are often not this well understood, but are nevertheless published. Once can always imagine "pathological" exotic cases where the unfolding is unreliable, but in such cases the parameterised approach used here (and in some cases even the full detector simulation and/or the detector calibrations!) would also be unreliable. (The desire of searches to use sparsely-populated bins does seem to be an intrinsic limitation on unfolding, however.)

We agree with this in principle, but the experience of those authors familiar with the inner procedures of ATLAS Standard Model measurements is that unfolding adds significant complexity and numerical stability issues to analyses, and as is proper in that situation, many requests are always made to prove the optimality of the unfolding step. And this is typically for a fairly well-understood phase-space. This feature is really aimed at BSM searches, which at least in the past few years were rarely unfolded both because of the extra time (even a small delay being enough to miss a key conference and get scooped), and because of the inapplicability of unfolding to low-population bins which may still have limit-setting power through Poisson statistics. We have modified the text to take these views into account, and to emphasise more strongly the relevance of low-population bins in searches.

In Section 2 the authors' approach is described as "a priori less accurate" than a DELPHES like detector model. I am actually convinced by the authors that in fact their approach is a priori more accurate in many cases, since the functions are tailored to specific analyses. Any generic detector simulation based on efficiency maps must surely contain compromises which will not be equally accurate for all event topologies?

We admit to having constructed something of a straw-man criticism here: we agree there is no intrinsic advantage to having an explicit detector+reco model if that model is not more accurate than transfer functions that could be measured and published. But the success of Delphes in particular (1300 citations and counting) is, in our experience, rooted in beliefs that its emulation of detector details makes it more accurate than a smearing approach. Our text was attempting to convince sceptical readers that this is not necessarily so, but could be improved: we have rephrased to made the "pedagogical" nature of this paragraph clearer.

The authors also state that "most calibrations are highly dependent on MC modelling". They should clarify what they mean by this. I don't think calibrations should be (or are) in the end highly dependent on the event generation, since they are validated using in situ measurements. However, they are dependent on the MC model of the detector (which is validated in various ways using data).

We will clarify this: the intention is not to say that calibrations are sensitive to MC generation details, but that MC simulations are often used to derive kinematic calibration corrections, using the MC truth as a target... and as mentioned here, the effects being corrected for are mostly those from the detector modelling rather than the MC event generation, which is usually good enough these days. As pointed out, these are also validated in situ. So the claim is too strong, and we are not sure that it can be also applied to efficiency calibrations: we have now weakened the statement, thanks.

In section 2, what justification is there for saying the 10-20% accuracy would lead to "conservative" bounds? Couldn't they just as well be over-aggressive, depending upon the sign of the error made?

This is very true: we were mixing up some issues here. The point is that exclusion contours are rather insensitive to yield-changes on this scale, so the discrepancy -- either conservative or overaggressive -- is relatively slight. We reworded.

In section 3 the authors mentuon "unconscious factors". I don't think unconscious is the right word here?

It was used to contrast with "explicit", so the obvious choice is "implicit". This seems a bit vague, so we have gone with "implicit, environmental factors".

Section 4.1 It seems odd, though probably justifiable, that Energy smearing is applied to momentum and that the mass is left unchanged. I presume the energy is recalculated so the result is a valid four vector? If the mass is to be then subsequently smeared, would the energy again be recalculated automatically? Some clarification would help, I think.

Yes, it can be done either way, and in fact the ability to provide the jet and particle smearing classes with an ordered list of smearer functions means that either first smearing in energy and then mass, or first in mass then in energy, can be done. This is noted in the text.

Section 4.2 "ghost-associated" needs description/reference

Done.

4.8 Jets have no efficiency calculated either? So it is not just MET...

Jets can have an efficiency, it is just not typically less than 1 and so we leave it as an optional argument when configuring the jet smearer (while the efficiency functor is a mandatory first argument for the particle smearer).

Section 5 was the 10GeV pT applied to truth or smeared value?

It was applied at whichever level the analysis was configured to run in: at truth-level for the Rivet analysis in "truth mode", and at reco-level for the Delphes analysis and the Rivet analysis in "reco mode". We clarified this in the text, and will also publish the analysis codes in the next arXiv update.

Best regards, Andy, Deepak, Karl

---

## Round 1 · Referee Report · Tilman Plehn (Referee 2) · 2019-10-30

Strengths

  • the authors present a new tool which improves the modelling of detector effects especially for subet physics;
  • the tool should be numerically efficient;
  • the description is nice and physics-oriented.

Weaknesses

  • see requested changes, nothing that cannot be fixed (or argued away)

Report

The paper has the potential to fill a known whole in LHC simulations, which too often rely on default Delphes, even though everybody knows that there are issues.

Requested changes

From the front to the back 1- define `ill-posed problem' as in unfolding. That accusation is a little too unspecific; 2- in the introduction it would be nice to mention that smearing is a very old way to describe detector effects. I learned this from Dieter Zeppenfeld in the late 90s, Tao Han had his famous hanlib with smearing functions. So while the presented approach is very useful, it is totally not new. Please make that clear and cite some old papers, for example Dieter's WBF papers should do. 3- I am sorry, but I do not get Fig.1. Why does the step from Det to Reco get us closer to MC? Is that guaranteed or hoped for? 4- global EFT analyses like we do them in SFitter are probably amongst the most sensitive users of detector simulation, and it's more actual physics than BSM stuff. Same for Glasgow's own TopFitter, just citing GAMBIT can be considered an insult here. 5- in 4.1, what about forward/tagging jets? 6- in 4.3 I do not understand what the authors are saying. 7- concerning 5, there is an ATLAS member on the paper and validation is all with Delphes, no data here? I am surprised about this lack of experimental honor! 8- for instance in Fig.2 the labels are too small to read on a laptop. 9- in Sec.6, the central limit theorem does not apply to profile likelihoods (as far as I understand), to that statement is a little pompous. 10- all over Sec.6 I am missing particle flow. Substructure tends to use lots of track information. At least comment and admit calorimeter defeat, please. 11- Eq.(2) is missing an error bar, so we can compare with Eq.(3). 12- in Sec.6.1, for instance, what is the effect of UE, pile-up, etc? 13- I learned that consistently writing in passive voice is bad style.

  • validity: top
  • significance: top
  • originality: ok
  • clarity: high
  • formatting: excellent
  • grammar: reasonable

Author:  Andy Buckley  on 2019-11-25  [id 657]

(in reply to Report 2 by Tilman Plehn on 2019-10-30)

Hi Tilman,

Thanks for the comments, very helpful. We'll fix the remaining issues and post an updated version as soon as possible, but wanted to respond here now rather than delay any further.

From the front to the back 1- define `ill-posed problem' as in unfolding. That accusation is a little too unspecific;

We are using ill-posed/well-posed in the standard statistical sense that it fails the Hadamard criteria, specifically the one about unique solutions. Is a reference needed?

2- in the introduction it would be nice to mention that smearing is a very old way to describe detector effects. I learned this from Dieter Zeppenfeld in the late 90s, Tao Han had his famous hanlib with smearing functions. So while the presented approach is very useful, it is totally not new. Please make that clear and cite some old papers, for example Dieter's WBF papers should do.

Absolutely: the intention is not to imply that it's new, but rather why we prefer the old, perceived inferior approach, when there's an "industry standard" explicit-geometry alternative.

3- I am sorry, but I do not get Fig.1. Why does the step from Det to Reco get us closer to MC? Is that guaranteed or hoped for?

The point of the Reco process is to reconstruct physics objects close to what was physically there, as opposed to the very distant Det-level data: either large sets of hits and clusters, or uncalibrated physics objects fully subject to detector biases (e.g. differences in EM vs hadronic calorimeter responses). If the Reco is not closer to the truth (i.e. MC) than the Det, then an awful lot of reco and calibration work has been for nothing.

4- global EFT analyses like we do them in SFitter are probably amongst the most sensitive users of detector simulation, and it's more actual physics than BSM stuff. Same for Glasgow's own TopFitter, just citing GAMBIT can be considered an insult here.

We're happy to add more citations here if appropriate, but note (self-interestedly!) that TopFitter only uses unfolded top measurements, i.e. ones where detector simulation does not enter. I had written this statement from the perspective of recasting fits using reco-level direct-search data, not EFTs where the devil is more in the details. For sure we should be also citing CheckMATE and maybe MasterCode/ATOM, and maybe a MadAnalysis interpretation paper; does SFitter really fit in the same category of a) using fast-sim, and b) not being very sensitive to the exact details?

5- in 4.1, what about forward/tagging jets?

We do not have a special treatment for forward jets, beyond what is given in public efficiency maps. Part of what we try to encourage with this development is the experiments providing their own custom response functions, in whatever detail they require, to allow reproduction of special phase-space such as forward jet reco.

6- in 4.3 I do not understand what the authors are saying.

We will expand this a little: it is vague!

7- concerning 5, there is an ATLAS member on the paper and validation is all with Delphes, no data here? I am surprised about this lack of experimental honor!

Ha! And there are two ATLAS authors on the paper! But there is very little data from either experiment that is published in a comparable way at both reco and unfolded levels. We note this in the substructure section where we do attempt tuning based on this reco/truth comparison, and could only find two suitable (partial) measurements... one of them produced on-demand in parallel with the evolution of this paper. In the conclusions we do motivate more such dual reporting, specifically to allow external development and validation of fast-sim codes.

8- for instance in Fig.2 the labels are too small to read on a laptop.

We will enlarge these, thanks.

9- in Sec.6, the central limit theorem does not apply to profile likelihoods (as far as I understand), to that statement is a little pompous.

I'm not sure where profile likelihoods come into this -- we're talking about smearing effects on kinematic observables. Does the point still stand?

10- all over Sec.6 I am missing particle flow. Substructure tends to use lots of track information. At least comment and admit calorimeter defeat, please.

Not all substructure uses track information: both the available cases with both reco and unfolded results, are ATLAS and pure calorimeter: there are no ATLAS public results with pflow jets, I think. But anyway this isn't an explicit fast-sim: the inspiration was that calo geometry would have some biasing effects, but in the more recent analysis using a cluster granularity grid everestimated the biases: reco algorithms really matter and can offset naive detector effects. The key thing is that any functional ansatz can be proposed and tested, and if it works it works, regardless of the detector components used.

11- Eq.(2) is missing an error bar, so we can compare with Eq.(3).

We will re-fit to get this information.

12- in Sec.6.1, for instance, what is the effect of UE, pile-up, etc?

UE is included in the MC model, pile-up is not. But anyway both should be largely cancelled by jet grooming, and whatever is residual is absorbed into the unfolding.

13- I learned that consistently writing in passive voice is bad style.

The standard, enforced style for the experiments: seems there's no community standard. But I suppose we can use the SciPost freedom of expression.

---

## Round 1 · Referee Report · Anonymous (Referee 3) · 2019-10-31

Strengths

1- The authors present an extension to the extremely widely used Rivet analysis tool to approximate the most important detector effects and include them in newly built analysis codes. 2- The structure and paradigms of the implementation are described in detail, and seem to be kept minimal, yet easily extendable.

Weaknesses

1- See requested changes below, in general only very minor clarifications are required.

Report

This paper greatly improves the current situation of the experiments publishing unusuable (or barely usable) data which is not corrected for proprietary and publicly unknown detector effects. Existing solutions such as Delphes have known issues and inaccuracies sometimes of the same size as the detector effects themselves. It is thus a very much appreciated service for the theory community and (I suppose) the LHC collaborations.

Requested changes

1- in how far is the problem of detector unfolding "ill-posed". In my understanding there should be clear, in the authors terms, forward transfer function that should be invertible, even if maybe multiple solutions exist. 2- first paragraph, end of last sentence, the "/" probably should be replace with ".". 3- I am not sure I fully understand Fig. 1. What is the difference and (even schematic) meaning between "Reco ??" and "Reco/analysis". Why are "Reco/analysis" objects further from "MC truth" than "Detector hits"? Please expand on the definitions used in that graph and its discussion. 4- Dublicated citation [7] to MadAnalysis. 5- Section 2, Implementation. In the first line "is" should probably be "in". Also this sentence needs some overhaul for logic. 6- Same section, bottom of the page: The "The SmearedJets ..." sentence is a bit convoluted and uses a superfluous semicolon. 7- Sec. 4.2, please defineghost-associated or provide a reference. 8- Sec. 4.2, should by any chance MV2c20 be MC2c20? 9- Sec. 5. Has any validation of the photon treatment been performed? 10- ".., with Rivet and Delphes seen to stick closely together ..". Could this be formulated less colloquially? 11- Sec. 6. The discussion of the substructure observables is carried out on the basis of calorimeter modelling, how about tracking information? Is tracking info modelled/used as well? 12- Sec. 6. The authors find that the fit of the parameters of their expected detector response varies greatly with the used data input. I would like to ask the authors to somewhat expand on their discussion of the implications of this finding, possibly also including that the expected functional form may be incorrect. As it stands, neither of the fitted parameter values inspires much confidence in actually being used in unmeasured regions/observables.

  • validity: high
  • significance: top
  • originality: good
  • clarity: high
  • formatting: excellent
  • grammar: reasonable

Author:  Andy Buckley  on 2019-11-25  [id 656]

(in reply to Report 3 on 2019-10-31)
Category:
answer to question

Thank you for the feedback. We've fixed the small issues, and will post an updated version with photon response comparisons as soon as possible, but wanted to respond here now rather than delay any further.

1- in how far is the problem of detector unfolding "ill-posed". In my understanding there should be clear, in the authors terms, forward transfer function that should be invertible, even if maybe multiple solutions exist.

I don't quite follow this statement: if there are multiple solutions, it is not invertible. We are using ill-posed/well-posed in the standard statistical sense that it fails the Hadamard criteria, specifically the one about unique solutions.

2- first paragraph, end of last sentence, the "/" probably should be replace with ".".

In Section 2: thanks, we will fix.

3- I am not sure I fully understand Fig. 1. What is the difference and (even schematic) meaning between "Reco ??" and "Reco/analysis". Why are "Reco/analysis" objects further from "MC truth" than "Detector hits"? Please expand on the definitions used in that graph and its discussion.

We will clarify this. The red endpoints of the lines are data stages and their separations are meaningful. The orange labels are "processes", and their exact position is not meaningful, just that they take place along each line between stages. The x and y directions have no meaning.

4- Dublicated citation [7] to MadAnalysis.

We will remove this, thanks.

5- Section 2, Implementation. In the first line "is" should probably be "in". Also this sentence needs some overhaul for logic.

Thanks.

6- Same section, bottom of the page: The "The SmearedJets ..." sentence is a bit convoluted and uses a superfluous semicolon.

Fixed.

7- Sec. 4.2, please defineghost-associated or provide a reference.

Definition and reference now provided.

8- Sec. 4.2, should by any chance MV2c20 be MC2c20?

No, the other way around: both are MV2s, standing for "MultiVariate". Thanks for spotting the inconsistency.

9- Sec. 5. Has any validation of the photon treatment been performed?

It has been checked, but is not shown here, mainly because our analysis sample does not have any prompt photons. We are generating an extra sample to add photon comparisons.

10- ".., with Rivet and Delphes seen to stick closely together ..". Could this be formulated less colloquially?

Now done -- we thought the colloquial made it more readable!

11- Sec. 6. The discussion of the substructure observables is carried out on the basis of calorimeter modelling, how about tracking information? Is tracking info modelled/used as well?

There is no explicit detector simulation, although it's true that our Run 1 ansatz for angular granularity was motivated by calorimeter cells. Essentially we only measure total reco performance, and if the smearing ansatz we use fits the available data then it's working well enough, whether or not tracking information was used in the real detector and real reco algorithms. We found the calorimeter ansatz worked well for this data (although the cell granularity was not compatible with Run 2 reco), which is from ATLAS and hence primarily calorimeter based. It would be interesting to try with CMS data and/or ATLAS pflow jets, but such data doesn't yet exist in dual reco/unfolded form.

12- Sec. 6. The authors find that the fit of the parameters of their expected detector response varies greatly with the used data input. I would like to ask the authors to somewhat expand on their discussion of the implications of this finding, possibly also including that the expected functional form may be incorrect. As it stands, neither of the fitted parameter values inspires much confidence in actually being used in unmeasured regions/observables.

Correct: we were simply demonstrating that it is possible to fit this phase-space with a simple ansatz, although there are many detailed processes influencing the substructure observables. More data would certainly be needed to constitute a global tuning of such functions... or the experiments could use simple forms like this to report effective smearing-function parametrisations on a per-analysis basis.

---

## Round 3 · Referee Report · Jonathan Butterworth (Referee 1) · 2019-12-30

Report

An improved version of an already good paper. My only (and entirely optional) comment/question is that now you have added citations to checkmate etc do you also want to add one to Contur, as an example of a BSM application which could directly make use of rivet routines that exploit your work?
  • validity: -
  • significance: -
  • originality: -
  • clarity: -
  • formatting: -
  • grammar: -

Author:  Andy Buckley  on 2020-01-24  [id 717]

(in reply to Report 1 by Jonathan Butterworth on 2019-12-30)

Thanks for the suggestion. We could do this, but it would be a bit pre-emptive as Contur has not yet made public use of reco-level analyses (although we are currently involved in this extension, e.g. via Les Houches projects). Since our reference of other BSM recasting tools was limited to those already have fast detector-emulation tools, and we did not reference other currently unfolded/particle-level fits which could use reco-level Rivet results, it would seem like special treatment and self-reference so we think it better to leave it as-is this time.

Anonymous on 2020-01-24  [id 719]

(in reply to Andy Buckley on 2020-01-24 [id 717])

Fair enough. All fine.

---

## Round 3 · Referee Report · Tilman Plehn (Referee 2) · 2020-1-20

Report

Thank you for taking into account my comments, all cool now!

---

## Round 3 · Referee Report · Anonymous (Referee 3) · 2020-1-22

Report

The revised version sufficiently addresses the raised points.

---

## Round 3 · Author Response

We present a resubmitted version, improved thanks to the referees' comments. Thanks again for your consideration.

---

## Round 3 · List of Changes

1. Extension of validation plots and discussion to include tau and photon observables, using ttgamma events.
  2. Inclusion of Rivet & Delphes analysis routines and Delphes steering cards, as ancillary material.
  3. Clarification of motivation issues like unfolding as an ill-posed problem, the discussion of smearing accuracy (and its geometric representation in Fig 1), definition of ghost association, and various suggested improvements to phrasing.
  4. Adding citations to CheckMATE and MadAnalysis5 in addition to Gambit, as active and public examples of recasting code using fast-simulated reco-level quantities (other suggested codes are non-public and/or use unfolded observables).
  5. Removal of poorly-defined error estimates on fitted substructure smearing variables: the numerical values are less important than the observation that a simple smearing ansatz like this can obtain good results. Undoubtedly, further focused development could improve further, but that is beyond the scope of this in-principle demonstrator.

---

## Editorial Decision

published